# Transcriptomic Insights into the Degradation Mechanisms of *Fomitopsis pinicola* and Its Host Preference for Coniferous over Broadleaf Deadwood

**DOI:** 10.3390/microorganisms13051006

**Published:** 2025-04-27

**Authors:** Jianbin Xue, Yulian Wei, Liting Chen, Haisheng Yuan

**Affiliations:** 1Key Laboratory of Forest Ecology and Management, Institute of Applied Ecology, Chinese Academy of Sciences, Shenyang 110016, China; xuejianbin22@mails.ucas.ac.cn (J.X.); chenliting24@mails.ucas.ac.cn (L.C.); 2University of Chinese Academy of Sciences, Beijing 100049, China

**Keywords:** *Fomitopsis pinicola*, brown rot, comparative transcriptome analysis, degradation discrepancy mechanism

## Abstract

The degradation of deadwood is a vital ecological process for geochemical cycling and biodiversity conservation, with two main routes of fungal degradation: brown and white rot. Brown rot fungi cause severe destruction of wood cellulose and lead to brown and modified lignin residue. *Fomitopsis pinicola* is a typical brown rot fungus with a distinct host preference for coniferous trees. The mechanisms through which this fungus degrades coniferous and broadleaf wood remain poorly understood. Therefore, in this study, a 60-day cultivation experiment involving *F. pinicola* growing on deadwood strips of *Pinus koraiensis* and *Betula platyphylla* separately was performed. A comparative transcriptome analysis was carried out to explore the mechanisms underlying the differences in degradation, in terms of both physicochemical properties and transcriptomic data. The findings revealed that the host preference of *F. pinicola* resulted in the more efficient degradation of coniferous wood than broadleaf wood, accompanied by higher gene expression levels. GO enrichment analysis indicated that this preference was primarily associated with the hydrolytic enzyme family and processes related to the Fenton reaction, which is characteristic of brown rot fungi. Furthermore, the KEGG pathways showed that the DEGs were enriched in mainly included histidine metabolism, fatty acid degradation, and so on, indicating underlying carbohydrate and lipid metabolism processes. These results support *P. pinicola*’s strong ability to degrade the deadwood lignin of *P. koraiensis*, reflecting its adaptive evolution in host selection and choice of different ecological niches.

## 1. Introduction

In forest ecosystems, deadwood serves as a crucial habitat and food source for numerous species [1,2], and a significant portion of accessible plant biomass is stored in standing dead trees and fallen wood [3]. Lignin, as an essential component of deadwood, is an aromatic polymer that is relatively difficult to degrade due to its complex and highly cross-linked structure [4]. In this context, wood-decaying fungi utilize oxidative enzymes and hydrolytic enzymes to decompose deadwood, thereby facilitating the cycling of vital nutrients such as carbon and nitrogen, driving material flows within the ecosystem [5,6,7]. Wood-decaying fungi can be classified into white, brown, and soft rot fungi based on the type of decay that they induce. Notably, white rot fungi have a stronger genetic capacity to encode lignocellulolytic enzymes compared to brown rot fungi [8]. Brown rot fungi, on the other hand, partially degrade lignin through the Fenton reaction, which involves the reduction of iron ions to ferrous ions, leading to the production of hydrogen peroxide. This non-enzymatic process is also regulated by oxalates and low-molecular-weight phenolic compounds [9,10,11].

The ecological niche width of a species is defined by tolerance curves that describe its changes in fitness across environmental gradients, which can provide essential theoretical guidance for the study of the adaptation of species to environmental factors and their ecological roles [12,13]. Wood-decaying fungi exhibit different adaptive decay mechanisms across various hosts, with some being generalists (thus distributing their fitness across multiple hosts) while others are specialists (with maximum fitness in specific environments). Krah et al. constructed an evolutionary model of host adaptation and found that white rot fungi more efficiently utilize angiosperms, leading to a stronger preference for these plants, while brown rot fungi are more restricted, resulting in a broader host range in gymnosperms [14]. Their ecological types range from specialists, which are limited to single tree species, to generalists, which are capable of utilizing both angiosperms and gymnosperms [15]. Consequently, brown rot fungi have become prominent targets of extensive genomic and transcriptomic studies.

*Fomitopsis pinicola* is a common brown rot fungus that is widely distributed in the forests of the northern hemisphere. It is particularly found on coniferous hosts and rarely reported on broadleaf trees [16,17], indicating a clear host preference. *Pinus koraiensis* and *Betula platyphylla* are the dominant tree species in different forest types of the Changbai Mountain Nature Reserve in Northeast China, being representative of the coniferous and broadleaf trees of temperate forests in Northern China, respectively [18,19]. Since the advent of RNA-seq transcriptomic analysis, it has become increasingly popular among researchers due to its accuracy and reproducibility in quantifying gene expression levels [20,21,22]. In recent years, an increasing number of researchers have applied comparative transcriptomics approaches in fungal studies [23,24,25,26].

To explore the host preference, degradation effectiveness, and underlying mechanisms of *F. pinicola*, *B. platyphylla* and *P. koraiensis* were selected as representative angiosperm and gymnosperm species, respectively. Through laboratory inoculation experiments and comparative transcriptomic analysis, the differences in the degradation of *B. platyphylla* and *P. koraiensis* by *F. pinicola* were investigated. In particular, we sought to answer the following questions: Can *F. pinicola* degrade angiosperm wood? If so, how do its degradation effects differ, and what are the underlying mechanisms? The answers to these questions are expected to contribute to our understanding of the selection mechanisms and ecological functions of *F. pinicola* in natural environments.

## 2. Materials and Methods

### 2.1. Degradation Experiment Setup

The pure strain of *F. pinicola* (No. IFP700) was isolated from fruiting bodies collected from fallen *Picea koranensis* deadwood in the Changbai Mountain Nature Reserve, China, on 12 July 2007. Starting on 20 January 2021, a 60-day indoor cultivation experiment was conducted with *F. pinicola* growing on wooden strips of *Pinus koraiensis* and *Betula platyphylla*. The mycelium culture medium consisted of 2% malt extract powder, 0.8% agar powder, and 1 L distilled water. The experiments were conducted in 800 mL hexagonal flasks, which were sealed with a plastic film. The wooden strips of *P. koraiensis* and *B. platyphylla* were cut from the outer heartwood into pieces measuring approximately 6 cm in length, 1 cm in width, and 1 cm in height. The wooden strips were then sterilized under high pressure (121 °C for 40 min) after soaking in water for 2 h. To achieve an even distribution of mycelium in the medium, the cultures were kept in a light-protected environment at 26 °C. After the mycelium had fully colonized the medium (within 7 days), the sterilized pine and birch strips were placed in culture flasks, with six strips per flask. The flasks were incubated at 26 °C for 60 days. A control group containing only the medium without wooden strips was also prepared, and the experiment was performed in triplicate.

### 2.2. Weight, Cellulose, and Lignin Loss of Wooden Strips After Degradation

After 60 days of incubation, the wooden strips with mycelium growth, as well as the mycelium from the pure culture medium, were removed from the culture flasks. The strips were cleaned, with the mycelium being scraped from the surface, dried to a constant weight at 60 °C, and then weighed. The weight loss of the wooden strips was calculated. The remaining wooden strips were ground into sawdust with a Bear FSJ-A03D1 grinder (Bear Electric Appliance Co., Ltd., Foshan, China) and sieved using a 40-mesh sieve. The sawdust was then used to measure the cellulose and lignin content, determined according to the improved method of Van Soest using a cellulose filter bag [27]. Mycelium samples grown on birch wooden strips were designated as FPB, those grown on pine wooden strips as FPP, and those grown on the pure medium as FPC.

### 2.3. RNA Extraction, Library Construction, and RNA-Seq

Total RNA was extracted from the three cultured hyphae (control and hyphae from the surfaces of birch and pine wooden strips separately) using Trizol reagent (Invitrogen Life Technologies, Carlsbad, CA, USA). The concentration, quality, and integrity of the RNA were assessed using a NanoDrop spectrophotometer (Thermo Scientific, Waltham, MA, USA). RNA samples (3 μg) were then used as input for RNA-seq library preparation. Sequencing libraries were generated using the TruSeq RNA Sample Preparation Kit (Illumina, San Diego, CA, USA). Briefly, mRNA was purified from the total RNA using poly-Toligo-attached magnetic beads. The RNA was fragmented in an Illumina proprietary fragmentation buffer containing divalent cations under a high temperature. First-strand cDNA was synthesized using random oligonucleotides and SuperScript II. Second-strand cDNA synthesis was carried out with DNA polymerase I and RNase H. The remaining protruding ends were converted to blunt ends using exonuclease/polymerase activity, followed by enzyme removal. The 3′ ends of the DNA fragments were adenylated, and Illumina PE adapters were ligated to prepare the fragments for hybridization. cDNA fragments of approximately 200 bp were selected using the AMPure XP system (Beckman Coulter, Beverly, CA, USA). After 15 PCR cycles, DNA fragments with ligated adapters at both ends were selectively enriched using the Illumina PCR Primer Cocktail. The quality and quantity of the final products were assessed using an Agilent 2100 Bioanalyzer (Agilent Technologies Inc., Santa Clara, CA, USA) with an Agilent High-Sensitivity DNA Analysis Kit. Sequencing was performed on the Hiseq platform (Illumina) by Paisenno Biotechnology Co., Ltd., China, with a sequencing depth of 6 G.

### 2.4. Data Processing and Visualization Analysis

The obtained sequencing data were stored in the NCBI Sequence Read Archive (SRA) database (PRJNA1202733). Raw data in FASTQ format were processed using Cutadapt (v1.15) to filter out low-quality sequences, resulting in clean data for further analysis. The reference genome and gene annotation file (GCA_000344655.2_Fompi3_genomic.fna) were downloaded from the NCBI for data mapping. The filtered reads were mapped to the reference genome using HISAT2 (http://ccb.jhu.edu/software/hisat2/index.shtml, accessed 7 January 2022), with a default mismatch tolerance of up to 2. The read alignment and distribution were calculated. HTSeq (0.9.1) was used to count the reads mapped to each gene, and the raw gene expression was determined. Gene expression was normalized using the fragments per kilobase of transcript per million mapped reads (FPKM). Differentially expressed genes were identified using DESeq (1.30.0), with log2FoldChange > 1 and *p*-value < 0.05 as criteria. Functional annotation was performed based on the Gene Ontology (GO) and Kyoto Encyclopedia of Genes and Genomes (KEGG) databases. Principal coordinate analysis (PCoA) was conducted based on the Euclidean distance. Statistical analyses were performed, including t-tests for cellulose loss and weight loss between FPP and FPB, and Wilcoxon rank-sum tests for lignin loss. All visualizations were created using R (version 4.3.2) with the ggplot2 and related packages.

## 3. Results

### 3.1. Wooden Strip Degradation

After 60 days of decomposition, the dry weight of the *B. platyphylla* wooden strips decreased by 49.33%, whereas that of the *P. koraiensis* wooden strips decreased by 53.38% (Figure 1A). The cellulose content of birch wood strips decreased by 32.98%, whereas that of pine wood strips decreased by 33.61% (Figure 1B); meanwhile, the lignin content of birch wood strips decreased by 26.88% and that of pine wood strips decreased by 32.82% (Figure 1C). The differences in dry weight and lignin content loss were significant (Figure 1A,C), whereas the difference in cellulose content loss was not significant (Figure 1B).

### 3.2. Data Quality Control Analysis of Transcriptome Sequencing

To understand the effects of different host wood types on the transcriptional levels in *F. pinicola*, RNA libraries of this species were built. After high-throughput sequencing, the libraries generated included 45,608,303 (FPB), 46,389,472 (FPP), and 45,487,677(FPC) clean reads—in particular, from paired-end reads with a single-read length of 150 bp and Q30 percentages (percentage of sequences with sequencing error rates < 1%) and mean GC content of 58%. These data indicate that the quality of the throughput sequencing was sufficiently high to warrant further analysis.

### 3.3. Analysis of Differential Expression of mRNA

The PCoA results for the three degradation groups showed significant differences among FPP, FPB, and FPC. The first two principal components explained 37.8% and 28.4% of the inter-group variation, respectively. Notably, the variation between replicates within the FPB group was more pronounced (Figure 2A). Compared to the control (FPC), the number of differentially expressed genes in *F. pinicola* decaying wooden strips of *P*. *koraiensis* was 2312, far more than that decaying wooden strips of *B*. *platyphylla*—especially regarding the upregulated genes (1083 vs. 511; Figure 2B). Among these DEGs, there were 1134 DEGs that were distinct with respect to the FPP–FPC comparison and 428 DEGs distinct with respect to the FPB–FPC comparison (Figure 2C). Compared to FPB, 213 genes were significantly upregulated and 516 genes were significantly downregulated in FPP (Figure 2D).

### 3.4. Functional Annotation of Differentially Expressed Genes

Enrichment analysis is mainly performed to test the degree of enrichment of certain functions or features in a gene set and to identify biologically meaningful patterns and functions from a large amount of genetic data. In particular, Gene Ontology Enrichment Analysis is conducted to examine the enrichment of Gene Ontology (GO) entries in a genetic dataset, which can help to understand the common characteristics of the associated genes in terms of biological processes (BP), molecular functions (MF), and cellular components (CC).

In the GO enrichment analysis, the differentially expressed genes from FPB wood were classified into 29 cellular components, 39 molecular functions, and 59 biological processes. The histogram of the top 40 GO enrichment terms with the lowest *p*-values, drawn after the classification of GO items, is shown in Figure 3A. Among them, GO terms relating to cellular components accounted for 67.5%. These genes are mainly related to metabolic processes, catalytic activity, the nucleus, intracellular membrane-bound organelles, and intracellular components, among others. When *F. pinicola* decayed wooden strips of *P. koraiensis*, the DEGs were classified into 22 cellular components, 46 molecular functions, and 52 biological processes. The histogram of the top 40 GO enrichment terms with the lowest *p*-values, drawn after the classification of GO items, is shown in Figure 3B. In contrast, most GO terms were related to molecular functions, accounting for 60%. The genes were related to catalytic activity, oxidoreductase activity, oxidation–reduction processes, carbohydrate metabolic processes, and integral components of the membrane, among others. Directed acyclic diagrams of the top 10 differing GO enrichment terms between *F. pinicola* decaying wooden strips of *P. koraiensis* and FPB with *p*-value < 0.05, drawn after the classification of GO items, are shown in Appendix A. The GO terms with the highest significance regarding biological process were oxidation–reduction processes, followed by isoprenoid metabolic and biosynthetic processes. The three most significant GO terms with respect to cellular components were related to the membrane, intrinsic components of the membrane, and integral components of the membrane. The significant GO terms relating to molecular functions included those associated with oxidoreductase activity, cation binding, cofactor binding, tetrapyrrole binding, metal ion binding, heme binding, and iron ion binding.

Furthermore, top 10 GO enrichment terms in relation to the number of DEGs between FPP and FPB with *p*-value < 0.05 are listed in Table 1. When *F. pinicola* decayed wooden strips of *P. koraiensis*, the number of upregulated genes was greater than when it decayed wooden strips of *B. platyphylla*. In the BP category, there were 255 differentially expressed genes related to oxidation–reduction, isoprenoid biosynthesis, transmembrane transport, and lipid biosynthetic processes, among others. In the CC category, 766 DEGs were related to the integral components of membranes (GO:0016021), intrinsic components of membranes, membrane, and cellular anatomical entity terms. Furthermore, there were 1034 DEGs classed into the MF category, which were found to be related to oxidoreductase activity (GO:0016491), iron ion binding, metal ion binding, catalytic activity, transition metal ion binding, and coenzyme binding, among others.

### 3.5. Analysis of Different Genes Using KEGG

The Kyoto Encyclopedia of Genes and Genomes (KEGG; https://www.kegg.jp/, accessed on 29 July 2023) pathways determine the main biochemical pathways and signal transduction pathways that candidate target genes are involved in, including the metabolism of carbohydrates, nucleotides, and amino acids and the biodegradation of organic matter. It not only provides all possible metabolic pathways but also comprehensively annotates the enzymes that catalyze each step of the reaction, including amino acid sequences, links to PDB libraries, and more. The top pathways with the lowest *p*-values (*p* < 0.05) are used to produce a KEGG enrichment bubble diagram, where the ordinate represents the pathway. Bluer and larger bubbles are used to indicate a larger number of differentially expressed genes enriched in a given pathway. In this study, the KEGG enrichment analysis results of differentially expressed genes are shown in the form of a scatter plot. The results of the FPB–FPC comparison indicated that there were four key pathways related to differentially expressed genes, which were related to multiple types of metabolism, such as ribosome biogenesis in eukaryotes, starch and sucrose metabolism, and so on (Figure 4A). Meanwhile, 19 metabolic pathways were enriched in the FPP–FPC comparison, mainly belonging amino acid metabolism, carbohydrate metabolism (including starch and sucrose metabolism), lipid metabolism, fatty acid degradation, pyruvate metabolism, glycerolipid metabolism, and so on (Figure 4B). Although starch and sucrose metabolism and methane metabolism were common metabolic pathways in the FPB–FPC and FPP–FPC comparisons, the associated differentially expressed genes were completely different.

## 4. Discussion

Previous research has shown that brown rot fungi prefer conifers in general and do not secrete peroxidases, instead utilizing the non-enzymatic Fenton reaction—that is, applying a special radical mechanism with hydroxyl radicals (HO-) as an oxidant species [28,29,30,31]. Moreover, brown-rot-decayed wood is dry, brown, powdery, and cracking to cubicles. These fungi contribute to improving the soil structure and carbon sequestration in the boreal forest. However, *F. pinicola* has also been found on angiosperm deadwood in the field, albeit only rarely [32,33]. Therefore, a question arises regarding the differences in the mechanisms with which *F. pinicola* degrades angiosperm wood in comparison with coniferous deadwood. In this context, a comparative transcriptome analysis allowed for exploration of the different mechanisms by which *F. pinicola* degrades coniferous and angiosperm deadwood and explained the reasons that *F. pinicola* is more inclined to select coniferous species as hosts naturally.

In the 60-day wooden strip degradation study, *F. pinicola* showed more effects on *P. koraiensis* than on *B. platyphylla*, with the dry weight loss being 53.38% versus 49.33%. In fact, deadwood of *B. platyphylla* always shows a higher degradation rate than deadwood of *P. koraiensis* in the field. Furthermore, the degradation rates of cellulose and hemicellulose in the wooden strips of *P. koraiensis* and *B. platyphylla* were similar, at about 33%, while the lignin content in birch and pine wooden strips decreased by 26.88% and 32.82%, respectively. These results show that the difference in the degradation rate between pine and birch deadwood is mainly reflected in the decomposition of lignin. However, previous research reported *P. koraiensis* to contain higher levels of lignin but smaller amounts of cellulose and hemicellulose when compared to *B. platyphylla* [34]. Thus, *F. pinicola* has a strong ability to degrade lignin from *P. koraiensis*.

According to the GO analysis results obtained for *F. pinicola* decaying wooden strips of *P. koraiensis* and *B. platyphylla* separately, molecular function categories relating to catalytic, hydrolase, oxidoreductase, and endopeptidase activity were all enriched, with low *p*-values, revealing that the mechanisms through which *F. pinicola* degrades cellulose in both coniferous and deciduous wood are similar. However, regarding lignin degradation, more GO terms belonging to molecular function categories were enriched in the FPP vs. FPC compared to the FPB vs. FPC comparison (e.g., iron ion binding, metal ion binding, transition metal ion binding, etc., which are related to lignin degradation). Furthermore, the number of differentially expressed genes was much higher in the FPP vs. FPC comparison (DEGs: 2312) than in FPB vs. FPC (DEGs: 1421). More brown-rot-related genes were differentially expressed in FPP, although some genes showed higher abundance in FPB. The functions related to brown rot are primarily involved in redox processes and the Fenton reaction [35]. This possibly explains why *F. pinicola* degrades pine wood more effectively, which is also consistent with Rehbein’s observation in a degradation experiment using larch wood and the brown rot fungus *Antrodia vaillantii* [36]. In addition, *Gloeophyllum neolentinus* and *Serpula hygrophoropsis* have been reported to specialize in degrading gymnosperms [14]. The transcriptomic analysis of *F. pinicola* during the degradation of *P. koraiensis* and *B. platyphylla* revealed that the mechanism underlying the preference of brown rot fungi for coniferous wood is related to the larger number of differentially expressed genes enriched in GO terms associated with lignin degradation molecular functions.

It has been previously reported that brown rot fungi do not secrete peroxidases [37]. This point may be debated given the presented transcriptomic analysis results for the typical brown rot fungus *F. pinicola* when decaying coniferous or deciduous deadwood. In our study, when *F. pinicola* degraded wooden strips of *P. koraiensis*, 11 differentially expressed genes were enriched in GO terms relating to peroxidase, glutathione peroxidase, and thioredoxin peroxidase activity. Notably, four differentially expressed genes were enriched in GO terms related to peroxidase activity when *F. pinicola* degraded wooden strips of *B. platyphylla*. In both *P. koraiensis* and *B. platyphylla*, *F. pinicola* exhibited shared characteristics regarding its carbohydrate-active enzymes (CAZymes), but specific differences were also observed. This finding aligns with the study of De Figueiredo [38], who investigated *Lentinus sulphureus* ATCC 52600 in terms of lignocellulose degradation. In a genome mining study of *F. pinicola*, Csarman identified potential laccase genes and observed weak reactions in plate assays when using poplar sawdust and ABTS as indicators of laccase activity [39]. Furthermore, Park et al. reported active laccase activity in a *F. pinicola* strain (Taxid: 743,788) [40]. In our study, we observed the enrichment of genes in GO:0005507 (copper ion binding) when comparing FPP and FPB. Based on these findings, we hypothesize that *F. pinicola* may have employed a similar mechanism in the present study, involving laccases and copper ion binding to facilitate deadwood degradation.

Metabolic pathways play crucial roles in organisms, being related to the conversion of nutrients into the required substances and energy, thus participating in the regulation of the growth, differentiation, and functions of cells. In the comparison with FPB, the KEGG analysis results indicated that many differentially expressed genes were enriched in these main three metabolic pathway groups at a higher level in FPP. The first group involves amino acid metabolism, including histidine metabolism; valine, leucine, and isoleucine degradation; valine, leucine, and isoleucine biosynthesis; and lysine degradation. Histidine metabolism plays an important role in the regulation of protein synthesis and enzyme activity, significantly impacting the fungal physiology and secondary metabolite production [41]. The degradation of valine, leucine, and isoleucine is essential for various metabolic processes, providing energy and sources of carbon for the synthesis of other amino acids. The second group is carbohydrate metabolism, including pyruvate metabolism, ascorbate and aldarate metabolism, glycolysis/gluconeogenesis, butanoate metabolism, glyoxylate and dicarboxylate metabolism, and pentose and glucuronate interconversions. Carbohydrate metabolism plays a key role in the degradation of wood, mainly involving the lignocellulose (cellulose, hemicellulose, and lignin) decomposition process by wood-decaying fungi. Among these pathways, pyruvate metabolism—as a key part of energy conversion—is vital for stress resistance and energy metabolism in fungi [42]. The third group is lipid metabolism, which plays a crucial role in the degradation of wood‌ through the breakdown of lignin, including fatty acid degradation, glycerolipid metabolism, the synthesis and degradation of ketone bodies, and steroid biosynthesis. From these KEGG pathways, the differentially expressed genes involved in the mechanisms driving the greater efficiency of *F. pinicola* in degrading deadwood of *P. koraiensis*—and, thus, its preference for coniferous trees—have been further validated [43].

The results of sporocarp-based taxonomic studies revealed that *P. pinicola* exhibits a stronger preference for coniferous deadwood. The wooden strip degradation experiment further demonstrated its higher degradation efficiency for *P. koraiensis* on pine wood, attributed to greater lignin degradation efficiency. The GO enrichment term and KEGG pathway analyses provide genetic-level evidence supporting *P. pinicola*’s strong ability to degrade deadwood lignin of *P. koraiensis*, reflecting its adaptive evolution in host selection. The preference of brown rot fungi for coniferous hosts is mainly manifested in their more efficient degradation of lignin in coniferous deadwood, occupying a different ecological niche compared to white rot fungi. With further advancements in scientific technology, we will be able to uncover new discoveries regarding wood-decaying fungi and their decomposition of deadwood using more sophisticated methods.

## Figures and Tables

**Figure 1 microorganisms-13-01006-f001:**
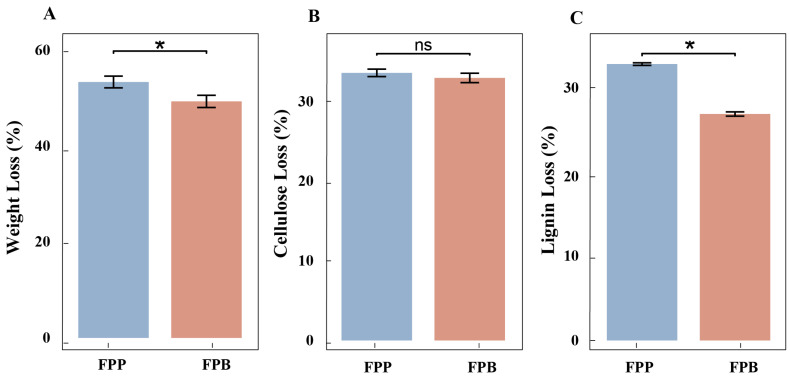
Changes in weight loss, cellulose loss, and lignin loss of wooden strips after 60 days of decomposition with *F. pinicola*. FPP represents wooden strips of *P. koraiensis* decomposed by *F. pinicola*, and FPB represents wooden strips of *B. platyphylla* decomposed by *F. pinicola*. (**A**) Weight loss of wooden strips. (**B**) Cellulose loss of wooden strips. (**C**) Lignin loss of wooden strips. Asterisks (*) show a significant difference between *F. pinicola*-degraded pinewood and *F. pinicola*-degraded birchwood (*p* < 0.05) and “ns” represents no significant difference.

**Figure 2 microorganisms-13-01006-f002:**
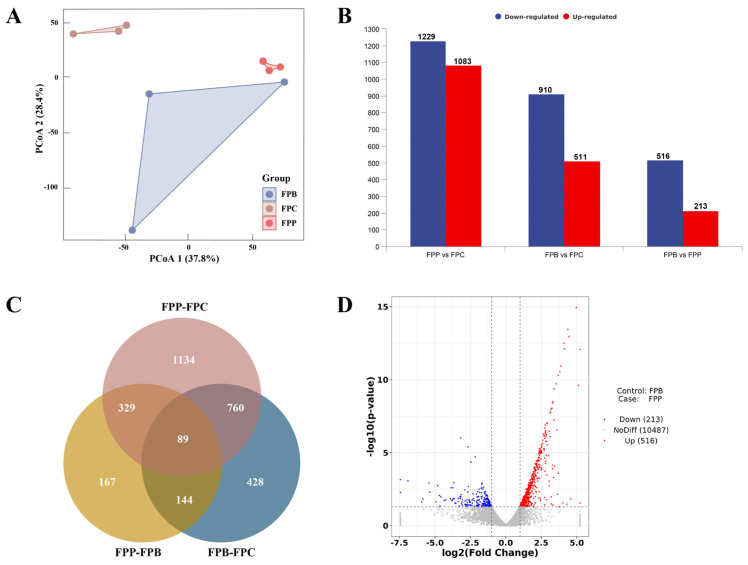
Overall analysis of differentially expressed genes in mycelium after 60 days of degradation across all groups. (**A**) Principal coordinate analysis (PCoA) of samples in the FPC, FPP, and FPB groups. (**B**) Number of differentially expressed genes in the FPP vs. FPC, FPB vs. FPC, and FPP vs. FPB group comparisons. (**C**) Venn diagram of differentially expressed genes (DEGs) in the transcriptome. The sum of the numbers in each circle represents the total number of DEGs in the comparison, and the overlap of the circles represents the common DEGs between the two groups compared. (**D**) A volcano map of the numbers of upregulated and downregulated genes in FPP compared with FPB (*p* < 0.05; log2(fold change) > 1 or <−1).

**Figure 3 microorganisms-13-01006-f003:**
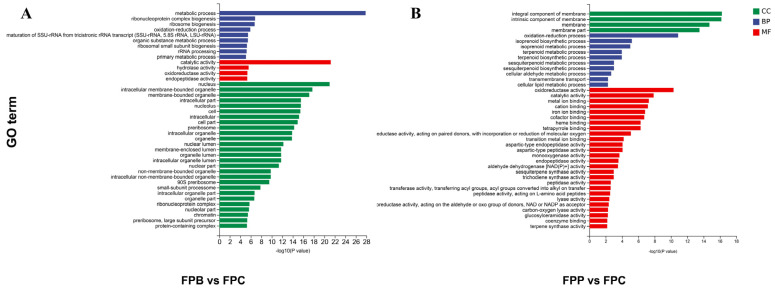
GO analyses of the differentially expressed genes in collected hyphae of *F. pinicola* following different degradation treatments. (**A**,**B**) Quantitative histograms of GO gene enrichment for FPB vs. FFC and FPP vs. FFC. MF: molecular function; BP: biological process; CC: cell component. The vertical axis represents the GO term, and the horizontal axis represents the −log10 (*p*-value) for GO term enrichment.

**Figure 4 microorganisms-13-01006-f004:**
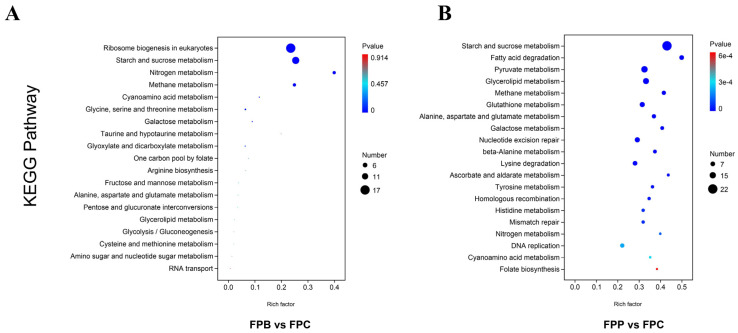
KEGG analyses of the differentially expressed genes in collected *F. pinicola* hyphae following different degradation treatments. (**A**) KEGG enrichment bubble charts of FPB vs. FPC comparison. (**B**) KEGG enrichment bubble charts for FPP vs. FPC comparison. The horizontal axis is the enrichment factor (expressed as the ratio of the number of differentially expressed genes annotated to the pathway/total number of expressed genes annotated to the pathway). The top pathways with the lowest *p*-values were used to produce the map, where the ordinate represents the pathway, the abscissa represents the enrichment factor (the number of differences in the pathway divided by all numbers), the size of each circle indicates the number of DEGs, and the color represents the significance (the bluer the color, the smaller the *p*-value). More differentially expressed genes are enriched in those pathways with bluer and larger circles.

**Table 1 microorganisms-13-01006-t001:** Top 10 GO enrichment terms and total numbers of DEGs in FPP compared with FPB (*p*-value < 0.05). For more GO enrichment terms, see Appendix A.

Category	GO Terms	Upregulated Genes	Downregulated Genes
BP	cellular aldehyde metabolic process cellular lipid metabolic process cellular response to toxic substance detoxification isoprenoid biosynthetic process isoprenoid metabolic process lipid biosynthetic process oxidation-reduction process response to toxic substance transmembrane transport	162	93
CC	cellular anatomical entity integral component of membrane intrinsic component of membrane membrane	528	238
MF	catalytic activity cation binding cofactor binding coenzyme binding heme binding iron ion binding metal ion binding oxidoreductase activity tetrapyrrole binding transition metal ion binding	684	350

## Data Availability

Data are contained within the article.

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
