# Peer review of "Transcriptomic Insights into the Degradation Mechanisms of Fomitopsis pinicola and Its Host Preference for Coniferous over Broadleaf Deadwood"

_microorganisms, 2025, doi:10.3390/microorganisms13051006_

Round 1
Reviewer 1 Report
Comments and Suggestions for Authors
In this manuscript transcriptomic analysis of F. pinicola growing on deadwood strips of Pinus koraiensis and Betula platyphylla was performed. Reviewer considers that this work has some importance in the field. However, in Reviewer opinion this work needs some major improvements before considering its publishing in Microorganisms (the manuscript contains too far reaching conclusions, some crucial information is missing, the goal is formulated incorrectly). English revision is required throughout the manuscript (sometimes awkward phrases or shortcuts are used). The conclusions are poorly formulated and does not summarize the results. In addition, the Reviewer has major concerns on the manuscript in terms of scientific content and writing (please see specific comments below).
Some key problems that should be addressed by the Authors are discussed below:
- Line 250-26: the sentence is rather laconic. In what sense the results explained the mechanism of wood decomposition? The mechanism of brown rot decay is already known. Please, be specific.
- Line 72-73: Is pinicola found growing on Betula in the Changbai Mountain Nature Reserve? In the paragraph above, the authors write that this fungus grows on angiosperms. So this question is unjustified.
- Line 90: How ‘full colonization of the medium’ was measured?
- Line 92: Why was cultivation in the dark chosen? In nature, neither trees nor fungi grow in the dark?
- Line 98: The idea behind wood degradation is that mycelium penetrate the wood. How do you know that the pieces of wood have been sufficiently cleaned?
- Line 103: Please, give more details on the applied method of Van Soest.
- Line 107: From which material was the RNA isolated? Please complete this.
- Line 153: I don't know if a 4% difference in weight loss can be considered significant after all?
- Figure 2B: Were there qualitative differences for up- and down- regulated genes for FPP, FPC, FPB?
- Table 1: The results in the table (GO ID) are not very informative. Perhaps they could be presented in a more accessible way?
- Line 319-321: When did the authors analyse CAZymes? CAZymes are involved in depolymerisation of lignin, cellulose, pectin and hemicellulose. I have not seen any such analysis, which should be performed for wood-degrading fungi, or references from the CAZY base. In addition, the authors were unable to observe enzyme secretion when carrying out the transcriptomic analysis. This is an overinterpretation.
- Line 327-330: This section is incomprehensible, contains factual errors and over-interpretations. laccase is not a peroxidase. The fact that laccase contains copper in its active centre does not mean that its function is to bind this metal this does not show that brown rot fungi can secrete peroxidases.
- Lines 364-368: this is not a conclusion
- Line 355-369: It is evident that the conclusions drawn are not reflected in the results obtained. I do not know how they reveal the mechanism of wood decomposition by pinicola and its host preference?
Minor corrections/comments:
- Line 82, italics,
- Line 325: active laccase activity’?
- Line 321: “Previous reported introduced brown-rot fungi did not secrete peroxidases”, need to be rephrased
Therefore, the Reviewer suggests manuscript major correction.
Comments on the Quality of English LanguageEnglish revision is required throughout the manuscript (sometimes awkward phrases or shortcuts are used).
Author Response
In this manuscript transcriptomic analysis of F. pinicola growing on deadwood strips of Pinus koraiensis and Betula platyphylla was performed. Reviewer considers that this work has some importance in the field. However, in Reviewer opinion this work needs some major improvements before considering its publishing in Microorganisms (the manuscript contains too far reaching conclusions, some crucial information is missing, the goal is formulated incorrectly). English revision is required throughout the manuscript (sometimes awkward phrases or shortcuts are used). The conclusions are poorly formulated and does not summarize the results. In addition, the Reviewer has major concerns on the manuscript in terms of scientific content and writing (please see specific comments below).
Reply: Thanks. We have promoted the conclusion and finished English language editing by MDPI authors services.
Some key problems that should be addressed by the Authors are discussed below:
- Line 25-26: the sentence is rather laconic. In what sense the results explained the mechanism of wood decomposition? The mechanism of brown rot decay is already known. Please, be specific.
Reply: Thanks for the suggestion, we have supplied the more details.
- Line 72-73: Is pinicolafound growing on Betula in the Changbai Mountain Nature Reserve? In the paragraph above, the authors write that this fungus grows on angiosperms. So this question is unjustified.
Reply: Thanks. There may be a misunderstanding. In line59-61, we described ”pinicola widely distributed in forest of the Northern Hemisphere. It is particularly found on coniferous hosts but rarely reported on broadleaf trees”. In this experiment, we used broad-leaved tree strips such as birch and red pine as experimental materials to analyze the mechanism differences in the degradation of broad-leaved and coniferous trees by pinicola, and attempted to analyze the reasons why pinicola is widely grown in coniferous trees rather than broad-leaved trees in the wild, which is also the main purpose of this study. And we didn’t illustrate F. pinicola grows on Betula in the Changbai Mountain Nature Reserve.
- Line 90: How ‘full colonization of the medium’ was measured?
Reply: In fact, this sentence means the mycelium colonized fully of the medium. The original sentence could easily cause ambiguity. The sentence has been modified.
- Line 92: Why was cultivation in the dark chosen? In nature, neither trees nor fungi grow in the dark?
Reply:There is an error about the cultivating description. However, there is a piece of glass on the door of the incubator that allows light to pass through. The sentence has been modified.
- Line 98: The idea behind wood degradation is that mycelium penetrate the wood. How do you know that the pieces of wood have been sufficiently cleaned?
Reply: The consideration about the mycelium in the wood is necessary. So we sterilized the wooden strips high pressure (121°C for 40 minutes) before them being placed in the culturing bottle.
- Line 103: Please, give more details on the applied method of Van Soest.
Reply: Thanks for the suggestion, we have supplied the more details.
- Line 107: From which material was the RNA isolated? Please complete this.
Reply: Thanks for the suggestion, we have supplied the more details.
- Line 153: I don't know if a 4% difference in weight loss can be considered significant after all?
Reply: The wooden strips were cut from fresh disc of living tree and it is hard to be decomposed at first phase. The 4% difference in weight loss is significant for the wooden strips with 60 days’ decaying.
- Figure 2B: Were there qualitative differences for up- and down- regulated genes for FPP, FPC, FPB?
Reply: Yes. Figure 2C verified this.
- Table 1: The results in the table (GO ID) are not very informative. Perhaps they could be presented in a more accessible way?
Reply: We added these GO terms as supplementary 2 to explain them.
- Line 319-321: When did the authors analyse CAZymes? CAZymes are involved in depolymerisation of lignin, cellulose, pectin and hemicellulose. I have not seen any such analysis, which should be performed for wood-degrading fungi, or references from the CAZY base. In addition, the authors were unable to observe enzyme secretion when carrying out the transcriptomic analysis. This is an overinterpretation.
Reply: As the wood-decaying fungus, some genes of F. pinicola belong to CAZymes. The CAZymes analysis in this manuscript is not the enzyme secretion but the gene expression. There were some description about the molecular functions of GO item in 3.4.
- Line 327-330: This section is incomprehensible, contains factual errors and over-interpretations. laccase is not a peroxidase. The fact that laccase contains copper in its active centre does not mean that its function is to bind this metal this does not show that brown rot fungi can secrete peroxidases.
Reply: In this paragraph, we discussed the the possibility about brown-decaying fungi secreting peroxidases based the 11 differential expressed genes were enriched into GO terms with peroxidase activity, glutathione peroxidase activity, thioredoxin peroxidase activity.
- Lines 364-368: this is not a conclusion
Reply: Thanks for this suggestion. It is not suitable and we have modified.
- Line 355-369: It is evident that the conclusions drawn are not reflected in the results obtained. I do not know how they reveal the mechanism of wood decomposition by pinicolaand its host preference?
Reply: Thanks for this suggestion. It is not suitable and we have modified.
Minor corrections/comments:
- Line 82, italics,
Reply: Thanks for this suggestion. We have modified.
- Line 325: active laccase activity’?
Reply: Thanks for this suggestion. We have modified.
- Line 321: “Previous reported introduced brown-rot fungi did not secrete peroxidases”, need to be rephrased
Reply: Thanks for this suggestion. We have modified.
Reviewer 2 Report
Comments and Suggestions for Authors
General comments:
Overall, the manuscript is well-written, concise, and readable. Introduction is well-written and has a readable flow, very concise. The experimental section was well-designed, and provided detailed information to reproduce the experiment. The findings are relevant to the scientific community, especially due to the ecological role of wood-decaying fungi. However, the discussion section needs improvement. The authors have good data that should be explored better in the discussion. The authors should answer the questions stated in the introduction, backed up by your data in the discussion. Finally, please, review the text, since there are some typos, non-itallized scientific names, and a few sentences that need to be restructured for clarity.
The manuscript is relevant for the field and it is suitable to publish after major revisions. I am looking forward to seeing the revised version.
Major revisions:
- Abstract
- Please review the last sentence of the abstract (lines 23-25), since it lacks clarity.
- I suggest to add a closing statement explaining how the results explained the preference of F. pinicola for coniferous deadwood and what is the impact of that for the scientific community. Is there an ecological relevance to your findings? Or is it a biotechnological relevance? The closing statement should give a take-home message based on the conclusions of your work.
2. Results
- Remove the 3.1.1 Subsubsection headline (line 147)
- Figure 1: add the y-axis legend for each graph.
- Figure 2: what the authors attribute to the variation between replicates in the FPB group?
- Figure 3C-E: acyclic diagrams are not readable. I suggest moving it to supplemental material, making it separated figures, with higher quality, so it is easy to read.
- Table 1: it will be more understandable for the reader if the authors add the name of each GO instead of the GO.ID, in the second column of Table 1. Just the GO.ID doesn’t say a lot unless the reader is very familiar with the ID.
- Line 252: change ‘were’ to ‘with’ in the following sentence: “.... four pathways related to differentially expressed genes were multiple types of metabolism…”
3. Discussion
- Overall, the discussion needs improvement. Several sentences need to be rewritten to improve clarity. In the discussion, the authors should address the questions stated in the introduction: “Could F.pinicola degrade angiosperm wood? If so, how does its degradation effects differ, and what are the mechanisms involved?” Also, the discussion should address with more detail what mechanisms your data suggests, and what is the ecological implications of that in the natural environment. The authors have good data that will support a better discussion.
- Lines 307-315: I suggest to re-written these two sentences. The message is not clear.
Author Response
Major revisions:
- Abstract
- Please review the last sentence of the abstract (lines 23-25), since it lacks clarity.
- I suggest to add a closing statement explaining how the results explained the preference of F. pinicola for coniferous deadwood and what is the impact of that for the scientific community. Is there an ecological relevance to your findings? Or is it a biotechnological relevance? The closing statement should give a take-home message based on the conclusions of your work.
- Reply: Thanks. We have promoted the conclusion
- Results
- Remove the 3.1.1 Subsubsection headline (line 147)
- Reply: Thanks for this suggestion. We have modified.
- Figure 1: add the y-axis legend for each graph.
- Reply: Thanks for this suggestion. We have modified.
- Figure 2: what the authors attribute to the variation between replicates in the FPB group?
- Reply: In field, F. pinicola always be found on pine deadwood while seldom on birch deadwood. And this species has been evolved a system suitable for the degradation of coniferous deadwood. So When F. pinicola degraded pine deadwood, the variation betweenreplicates was very small. But birch is not the natural host of F. pinicola, it was guessed that the DEGs will be more random and the variation between replicates was very big when F. pinicola degraded birch wood.
- Figure 3C-E: acyclic diagrams are not readable. I suggest moving it to supplemental material, making it separated figures, with higher quality, so it is easy to read.
- Reply: We moved Figure 3C-Eto supplementary 1 to show them clearly.
- Table 1: it will be more understandable for the reader if the authors add the name of each GO instead of the GO.ID, in the second column of Table 1. Just the GO.ID doesn’t say a lot unless the reader is very familiar with the ID.
Reply: Thanks for the suggestion. We added these GO ID terms as the supplemental 2.
- Line 252: change ‘were’ to ‘with’ in the following sentence: “.... four pathways related to differentially expressed genes were multiple types of metabolism…”
- Reply: Thanks for this suggestion. We have modified.
- Discussion
- Overall, the discussion needs improvement. Several sentences need to be rewritten to improve clarity. In the discussion, the authors should address the questions stated in the introduction: “Could F.pinicola degrade angiosperm wood? If so, how does its degradation effects differ, and what are the mechanisms involved?” Also, the discussion should address with more detail what mechanisms your data suggests, and what is the ecological implications of that in the natural environment. The authors have good data that will support a better discussion.
- Reply: Thanks for this suggestion. We have modified.
- Lines 307-315: I suggest to re-written these two sentences. The message is not clear.
- Reply: Thanks for this suggestion. We have modified.
Round 2
Reviewer 2 Report
Comments and Suggestions for Authors
I appreciate that the authors took the time to address the comments and have improved the manuscript overall. Most of my comments were addressed appropriately; however, a few important revisions are still needed:
-
Abstract – Closing Statement: The final sentence of the abstract still lacks a clear statement on the broader significance of the findings. What are the ecological or biotechnological implications of this work? The reader expects to see the relevance and potential applications of the study clearly articulated in the abstract.
-
Table 1: Presenting only the GO IDs in Table 1 is insufficiently informative. GO IDs alone do not convey meaningful insight, making the counts of up- and down-regulated genes difficult to interpret. I appreciate the addition of supplementary material with more details. However, the table would benefit from including GO term names directly or, at the very least, highlighting only the GO terms that are most relevant to the key findings or mechanisms involved in the decomposition pathway.
-
Discussion: The discussion section still requires substantial improvement. It does not fully explore the significance of the findings. The authors have compelling data that could support a more thorough and insightful discussion, particularly by connecting the results back to the questions posed in the introduction. Moreover, the ecological and biotechnological implications of the study remain underdeveloped and should be more clearly emphasized.
Author Response
Abstract – Closing Statement: The final sentence of the abstract still lacks a clear statement on the broader significance of the findings. What are the ecological or biotechnological implications of this work? The reader expects to see the relevance and potential applications of the study clearly articulated in the abstract.
Reply: The final sentence is modified: These results supported P. pinicola’s strong ability to degrade deadwood lignin of P. koraiensis, reflecting its adaptive evolution in host selection and choice of different ecological niche.
Table 1: Presenting only the GO IDs in Table 1 is insufficiently informative. GO IDs alone do not convey meaningful insight, making the counts of up- and down-regulated genes difficult to interpret. I appreciate the addition of supplementary material with more details. However, the table would benefit from including GO term names directly or, at the very least, highlighting only the GO terms that are most relevant to the key findings or mechanisms involved in the decomposition pathway.
Reply: We selected the top 10 of BP and MF category to show in the table and the total GO enrichment terms were attached in the supplementary file.
Discussion: The discussion section still requires substantial improvement. It does not fully explore the significance of the findings. The authors have compelling data that could support a more thorough and insightful discussion, particularly by connecting the results back to the questions posed in the introduction. Moreover, the ecological and biotechnological implications of the study remain underdeveloped and should be more clearly emphasized.
Reply: We have modified discussion according the good suggestion. These sentences were marked with blue.
